# AN ITERATIVE PROMPTING FRAMEWORK FOR LLM-BASED DATA PREPROCESSING

## ABSTRACT

Data preprocessing plays a crucial role in machine learning, directly impacting model convergence and generalization, especially for simple yet widely used linear models. However, preprocessing methods are diverse, and there are no deterministic rules for selecting the most suitable method for each feature in a dataset. As a result, practitioners often rely on exhaustive manual searches, which are both time-consuming and costly. In this paper, we propose an LLM-based iterative prompting framework that automates the selection of preprocessing methods. Our approach significantly reduces the number of iterations required to identify effective preprocessing strategies, thereby lowering human effort. We conduct an ablation study to analyze the contribution of each design component and provide extensive empirical evaluations. Results show that our method matches or surpasses baselines while substantially improving efficiency. The discovered preprocessing methods also accelerate training—either by improving convergence speed, enhancing generalization performance, or both.

## 1 INTRODUCTION

Data preprocessing is a critical stage in the machine learning (ML) pipeline: it shapes the optimization landscape (e.g., conditioning) and impacts generalization through bias–variance tradeoffs (Maharana et al., 2022; de Amorim et al., 2023; Li et al., 2017). In practice, however, selecting effective, leakage-safe[1] transformations for heterogeneous features is labor-intensive and often dominates project time. This has motivated research on automating preprocessing within end-to-end ML systems.

AutoML methods treat preprocessing as part of a pipeline search problem, typically combining algorithm selection with hyperparameter tuning via Bayesian optimization or evolutionary search (e.g., Auto-WEKA (Thornton et al., 2013), auto-sklearn (Feurer et al., 2015), TPOT (Olson et al., 2016)). These systems explore curated libraries of preprocessing operators and models under cross-validated evaluation (He et al., 2021; Truong et al., 2019). Reinforcement-learning-based approaches go further by modeling pipeline construction as a sequential decision process over editable primitives (Drori et al., 2018). More recently, LLM-based systems have been used to synthesize data-wrangling code and propose preprocessing steps (Hong et al., 2025; Guo et al., 2024) (Zhang et al., 2024b; Meguellati et al., 2025), sometimes coupled with programmatic prompt-optimization frameworks that iterate over feedback to refine suggestions (Li et al., 2025; Qi et al., 2024).

Despite progress, three challenges persist. (i) Search cost and rigidity: AutoML systems often operate over fixed, hand-curated operator sets; exhaustive or near-exhaustive exploration becomes computationally expensive as feature-dependent choices and operator hyperparameters expand combinatorially (Mumuni & Mumuni, 2025). (ii) Sample and compute inefficiency: RL-based pipeline synthesis improves automation but typically requires many environment interactions, careful reward shaping, and substantial compute to achieve stable policies (Yang et al., 2021; Cai et al., 2023). (iii) LLM fragility and objective mismatch: LLM-driven preprocessing can suffer from prompt sensitivity, hallucinated transformations, and misalignment with the low-epoch validation objectives

---

[1]This term means that no information from the validation or test sets is allowed to influence the fitted preprocessing or the model during training. All statistics, choices, and fitted parameters must be learned only from the training data of the current split, then applied to the held-out data purely for evaluation.

practitioners actually care about (Madaan et al., 2023; Zhang et al., 2024a; Narayan et al., 2022); moreover, enforcing leakage-safe evaluation and reproducibility remains nontrivial. These limitations motivate approaches that are feature-aware, budgeted (few-epoch), and evaluation-grounded rather than purely exploratory.

We introduce an LLM-based iterative prompting framework for preprocessing that searches over input/target mappings using a fixed-iteration budget and hold-out validation as a proxy objective. Concretely, for a given budget $k$, the system proposes candidate preprocessing strategies, trains the downstream model for exactly $k$ epochs on transformed training data, and selects the strategy that minimizes validation loss on transformed validation data—yielding a search that is aligned with practical time/compute constraints while remaining feature-dependent and leakage-safe.

The remainder of this paper is organized as follows. We begin by introducing the basic background of statistical learning and linear regression, followed by mathematical insights into why preprocessing is critical for learning performance, affecting both convergence and generalization. Next, we present our objective formulation and the proposed LLM-based approach for selecting preprocessing strategies. We then provide an extensive ablation study to validate each design choice, and finally compare our approach against baselines on a variety of benchmark datasets to validate its efficacy.

## 2 BACKGROUND AND PROBLEM SETTING

This section briefly reviews the basic background of the statistical learning setting and explains why data preprocessing matters for both convergence rate and generalization. We then introduce our learning objective, which aims to optimize the selection of preprocessing strategies through an LLM-based prompting framework.

### 2.1 STATISTICAL LEARNING BACKGROUND: WHY PREPROCESSING MATTERS

Let $X \in \mathbb{R}^{n \times d}$ and $y \in \mathbb{R}^n$. Conventional statistical learning setting typically assumes that $X, y$ are sampled from some unknown probability distribution. A fundamental algorithm is linear regression, that attempts to capture the linear relationship between the input feature $X$ and output target $y$. Such basic setting could help understand why data preprocessing matters. We refer to A.1 for more mathematical details regarding why preprocessing might affect convergence and generalization.

**Convergence perspective.** Consider the empirical square-loss, along with the gradient and Hessian notations:

$$f(w) = \frac{1}{2n}\|Xw - y\|_2^2, \qquad \nabla f(w) = \frac{1}{n}X^\top(Xw - y), \qquad H := \nabla^2 f(w) = \frac{1}{n}X^\top X.$$

It is known that the convergence is governed by the spectrum of $H$. Specifically, the condition number: $\kappa := \frac{\lambda_{\max}(H)}{\lambda_{\min}(H)}$ (Nocedal & Wright, 2006; Karimi et al., 2016).

Preprocessing can substantially improve the numerical properties of the learning problem by reducing the condition number of the data covariance matrix, which in turn accelerates convergence of gradient-based optimization (Gutman & Peña, 2021). Intuitively, good preprocessing makes the problem more "well-conditioned," meaning the optimizer can move towards the solution with fewer small or unstable steps.

For example, scaling or standardizing features (e.g., z-score normalization) ensures that all features are on a comparable scale, preventing a single high-variance feature from dominating the optimization dynamics. Similarly, whitening or decorrelation methods such as PCA transform the data so that features are uncorrelated and of equal variance; in this ideal case, gradient descent can converge to the solution in just one iteration. Finally, centering the data (subtracting the mean from both features and targets) removes the need for the optimizer to handle large intercept terms separately, further improving numerical stability.

**Generalization perspective.** Under the well-specified model $y = Xw^\star + \varepsilon$ with $\mathbb{E}[\varepsilon] = 0$ and $\mathrm{Var}(\varepsilon) = \sigma^2 I$, we have $\hat{w}_{\mathrm{OLS}} = (X^\top X)^{-1}X^\top y$, $\quad \mathrm{Var}(\hat{w}_{\mathrm{OLS}}) = \sigma^2(X^\top X)^{-1}$. The prediction variance at test point $x$ is $\sigma^2 x^\top(X^\top X)^{-1}x$ (Hastie et al., 2009). Preprocessing that inflates small eigenvalues of $X^\top X$ or removes near-null directions (e.g., via PCA truncation) reduces variance and can lower test MSE (Hoerl & Kennard, 1970).

When preprocessing is an invertible linear rescaling (e.g., standardization) applied consistently to train and test data, OLS predictions are unchanged—only the parameterization differs. Generalization is affected when regularization or early stopping are used, or when the transform is non-invertible (e.g., PCA, binning). In ridge and Lasso, unstandardized features receive uneven penalties, biasing estimates and hurting test error (Hastie et al., 2009; Hoerl & Kennard, 1970); standardization balances penalties and improves performance. Early stopping similarly acts as directional shrinkage (Ali et al., 2019; Sonthalia et al., 2024), and preprocessing that flattens the spectrum (scaling, whitening, PCA) makes this implicit regularization more uniform, leading to better generalization.

While spectral analysis of linear regression provides intuition for why preprocessing affects both optimization and generalization, the practical task of selecting preprocessing strategies is far from straightforward. Defining a precise objective—such as minimizing the condition number or balancing the bias–variance tradeoff—is difficult, since the space of possible transformations is complex, not explicitly enumerable, and often computationally infeasible to search exhaustively. Moreover, the design space of column-wise and joint transformations, along with their hyperparameters, is combinatorial. Each candidate pipeline requires evaluation through cross-validation under leakage-safe protocols, and the resulting implementations add maintenance overhead as data distributions evolve.

Industry surveys report that roughly $40\%$–$60\%$ of practitioners' time is spent on data preparation tasks rather than modeling (QuantumBlack, 2020; TMMData & Association, 2017), making preprocessing both time-consuming and human-intensive. We therefore explore an LLM-driven approach that proposes and refines preprocessing pipelines via iterative feedback, aiming to reduce human effort without sacrificing convergence or generalization.

## 2.2 OBJECTIVE OF DATA PREPROCESSING

We formalize preprocessing as a pair of mappings $r_x$ and $r_y$ that transform inputs and targets, respectively. Let $f^{(k)}$ denote a predictor obtained after $k$ training iterations (e.g., gradient steps) on the transformed data. Our ideal goal is a two-layer optimization:

$$\min_{r_x, r_y} \ \min_{f, k} \ \mathbb{E}\big[\, \ell\big(f^{(k)}(r_x(X)), \ r_y(Y)\big)\,\big], \tag{1}$$

which seeks preprocessing strategies $(r_x, r_y)$ that minimize the best attainable generalization loss, achieved in as few training iterations $k$ as possible. This objective is not directly tractable: the search space over $(r_x, r_y)$ is intricate and not explicitly enumerable, and the inner optimization over $(f, k)$ is computationally prohibitive.

Consequently, we adopt a practical proxy. We approximate the expectation with a hold-out validation error and treat $k$ as a budgeted hyperparameter (fixing the number of training iterations/epochs). Let $\mathcal{T}$ and $\mathcal{V}$ be a train/validation split. For any candidate $(r_x, r_y)$, we (i) fit $r_x, r_y$ on $\mathcal{T}$ and apply them to both $\mathcal{T}$ and $\mathcal{V}$ (to avoid leakage), (ii) train the model for exactly $k$ iterations on the transformed $\mathcal{T}$, and (iii) evaluate the validation loss on the transformed $\mathcal{V}$. The resulting implementable objective is

$$\min_{r_x, r_y} \ \widehat{\mathbb{E}}_{(x,y)\in\mathcal{V}}\Big[\, \ell\big(f^{(k)}_{r_x, r_y}\big(r_x(x)\big), \ r_y(y)\big)\Big], \tag{2}$$

where $f^{(k)}_{r_x, r_y}$ is the predictor obtained after $k$ iterations of training on $\big(r_x(\mathcal{T}_X), \ r_y(\mathcal{T}_Y)\big)$. In our framework, an LLM proposes candidate $(r_x, r_y)$ strategies, and we select the one that minimizes the above proxy objective under the fixed iteration budget $k$.

# 3 LLM-BASED ITERATIVE APPROACH: LLM-PRESTO

This section introduces our main approach, LLM-Presto (LLM-based Preprocessing Strategy Optimization), for optimizing preprocessing methods, followed by variants that serve as baselines or ablation settings in the experimental section.

## 3.1 OUR APPROACH: LLM-PRESTO

Our procedure performs an iterative, LLM-driven search that approximately solves the implementable version of the ideal two-layer optimization stated previously.

**Step 1**: Preprocessing strategy generation. An LLM is prompted with a summary of the dataset and task (feature counts and types, basic statistics, target type), the downstream model class and hyperparameters, and the iteration budget $k$. Conditioned on this context, the LLM proposes a preprocessing strategy $(r_x^{(t)}, r_y^{(t)})$ at iteration $t$, comprising concrete input/target transformations (e.g., scaling, imputation, encoding, PCA, target processing).

**Step 2**: Leakage-safe evaluation under a fixed budget. For the proposed $(r_x^{(t)}, r_y^{(t)})$:

1. Fit transforms on train only: estimate all parameters of $r_x^{(t)}$ and $r_y^{(t)}$ using $\mathcal{T}$; apply the fitted transforms to both $\mathcal{T}$ and $\mathcal{V}$. 2. Train for $k$ iterations: train the downstream model on $r_x^{(t)}(\mathcal{T}_X)$ for exactly $k$ iterations to obtain $f_{r_x^{(t)}, r_y^{(t)}}^{(k)}$. 3. Score the strategy: compute $L_t := \widehat{\mathbb{E}}_{(x,y) \in \mathcal{V}} \left[ \ell \big( f_{r_x^{(t)}, r_y^{(t)}}^{(k)} (r_x^{(t)}(x)), \ r_y^{(t)}(y) \big) \right]$. Record $L_t$ and maintain the incumbent best $L^\star = \min_{s \le t} L_s$ with corresponding $(r_x^\star, r_y^\star)$.

**Step 3**: Iterative refinement via feedback. Provide the LLM with structured feedback from Step 2 (e.g., training/validation losses or task-appropriate metrics, the incumbent $L^\star$, and a brief summary of what changed in $(r_x^{(t)}, r_y^{(t)})$). Using this feedback, the LLM proposes a refined strategy $(r_x^{(t+1)}, r_y^{(t+1)})$. Iterate Steps 1–3 until a stopping rule is met: (i) the LLM signals no further improvements, (ii) $L^\star$ has not improved for a preset number of iterations, or (iii) a compute/prompt budget is reached.

Return the best strategy found, $(r_x^\star, r_y^\star) \in \arg \min_t L_t$, i.e., the empirical minimizer explored by the loop. This procedure therefore optimizes the proxy objective derived from equation 1 by (a) fixing the iteration budget $k$, (b) evaluating generalization via hold-out validation, and (c) using an LLM to generate and refine candidate $(r_x, r_y)$ in a leakage-safe, feedback-driven search.

This iterative feedback loop allows the LLM to explore a wide preprocessing space while remaining computationally feasible.

**Potential Variants of LLM-Presto.** To better understand the contribution of each component in our approach, we define several variants below.

**Zero-shot single-pass LLM:** This approach queries the LLM with the same initial prompt and generates the preprocessing strategy once, without any iteration or feedback. It serves as a minimal baseline for evaluating whether iterative refinement indeed provides a better strategy than a one-time request.

**Self-refine (iterative) (Madaan et al., 2023):** This baseline is designed as repeatedly prompting the LLM to critique and improve upon its latest strategy through multiple rounds. The process continues until the LLM explicitly outputs "no changes". This extension explores whether iterative self-refinement yields better preprocessing strategies compared to both single-query refinement and external iterative feedback.

**Self-refine (single query):** Inspired by the iterative Self-Refine approach, we design a baseline that instructs the LLM to critique its own suggestions and propose an improved version in response. This can reduce the need for extra external iterations and can generate better preprocessing strategies in one interaction. This strategy focuses on exploring whether self-improvement can replace external model training and feedback. It serves as a baseline that provides a refinement mechanism without requiring interaction, allowing us to assess the benefit of an explicit interaction.

In the following discussion, we refer to self-refine (single query) as **Self-refine I** and self-refine (iterative) as **Self-refine II**.

**Chain-of-thought (CoT) (Wei et al., 2022):** The LLM is instructed to generate preprocessing strategies with reasoning step-by-step. CoT helps the LLM to emphasize the rationale behind each transformation, which can lead to the production of more logical strategies and can improve interpretability. It may also reduce the likelihood of producing unjustified preprocessing strategies. Since CoT is a widely adopted prompting technique, we use it as a baseline to benchmark our iterative feedback approach against a standard reasoning-based paradigm.

These variants allow us to systematically investigate how iterative refinement and prompting style affect preprocessing strategy search. This provides a clear basis for evaluating the specific advantages of our proposed approach.

## 4 EMPIRICAL STUDIES

Our experiments are divided into two parts. The first part presents an ablation study that examines the contribution of each design choice in our proposed method on California Housing (CAH) dataset. The second part reports the overall empirical results on additional five benchmark datasets. The datasets used are: for classification tasks: Adult Census Income (ADU), Obesity Risk (OBS), and Higgs Boson (HIG); for regression tasks: Wine Quality-White (WQW), California Housing (CAH), and Ames Housing (AMH). These datasets are frequently used in related works (Gijsbers et al., 2024; Li et al., 2023). Details of the Datasets can be found in Table 8 in Appendix A.3.

### 4.1 ABLATION STUDY

The ablation study is designed with two main objectives: 1) Assessing the effect of using a limited number of epochs (i.e. the fixed budget in **Step 2** of our approach) to evaluate the performance of different preprocessing methods, and 2) investigating the impact of incorporating various forms of feedback into prompts when requesting preprocessing suggestions. The experiments are conducted on the California Housing Dataset.

**The effect of fixed budget**. We investigate how the number of searching epochs k in the feedback affects both the efficiency and stability of finding the optimal preprocessing strategy. Table 1 shows how $k$ affects the efficiency and stability of finding the best preprocessing method. Very small budgets ($k = 1$) are unstable and often lead to poor strategies, while larger budgets ($k \geq 10$) provide little improvement but incur substantially higher costs. Across learning rates, $k = 5$ consistently achieves the best trade-off, matching or outperforming a larger $k$ in validation loss while requiring 30–50% fewer epochs. We also note that $k = 20$ occasionally produces anomalous results, where the LLM shifts focus toward feature compression (e.g., PCA) prematurely, leading to unexpected high validation losses. A detailed discussion of this phenomenon is included in the Appendix A.2. We therefore adopt $k = 5$ as the default in subsequent experiments.

Table 1: Best loss and total epochs of different feedback budgets and learning rates. $k$ stands for the number of searching epochs included in the feedback prompt. LR stands for the learning rate. In this table, total epochs include the epochs during searching process. Best Loss is the smallest validation loss achieved. Experiments are conducted on the California Housing Dataset, with learning rate starting at 0.005, which is found to be optimal for the unpreprocessed dataset through parameter sweep experiments.

| LR | 0.005 | | 0.01 | | 0.1 | |
| $k$ | Best Loss | Total Epochs | Best Loss | Total Epochs | Best Loss | Total Epochs |
|---|---|---|---|---|---|---|
| 1 | 0.6762 | 32 | 0.6223 | 22 | 0.6409 | 24 |
| 5 | 0.5970 | 88 | 0.5669 | 67 | 0.5767 | 54 |
| 10 | 0.6137 | 106 | 0.5605 | 124 | 0.5772 | 126 |
| 20 | 0.7127 | 277 | 0.7552 | 552 | 0.5708 | 209 |

**Investigate the utility of prompt content**. With this experiment, our aim is to determine which information should be included in the feedback prompt of **Step 3**. We test different signals: **Sample data**, a subset of the training set that exposes feature semantics important for imputation or encoding; **Unprocessed training losses**, i.e., raw training and validation losses with model parameters (weights and bias), which directly reflect optimization dynamics; **Training losses from the previous round**, which help the LLM refine its recommendations based on prior outcomes; **Model parameters**, namely weights and bias with feature names, which capture feature importance and guide dataset-specific strategies; **Aggregated statistics**, such as the best validation loss so far, which summarize progress across rounds; and **Derived signals**, including higher-order indicators like gradient norms. For convex models such as linear regression, training and validation losses are typically sufficient, while exploration of higher-order signals is left to future work.

Table 2: Final validation loss with different prompting contents. The first two variants are evaluated on the zero-shot result, as they directly affect the zero-shot performance. The remaining variants are evaluated one by one incrementally with our proposed method, LLM-Presto.

| | Final Loss | |
|---|---|---|
| | **with** | **without** |
| **Sample Data from Training Set** | 0.7052 | 0.7724 |
| **Unpreprocessed Training Losses for Each Epoch** | 0.7378 | 0.7052 |
| **Training Losses for Each Epoch of Last Round** | 0.6295 | 0.6673 |
| **Model Parameters** | 0.5868 | 0.6295 |
| **Best Validation Loss so far** | 0.5669 | 0.5868 |

Table 2 reports the final validation loss when including or excluding each prompt component. **sample data** clearly improves results by exposing feature semantics and aiding imputation, especially with missing values. In contrast, prompting with **unprocessed training losses** does not help and can even hurt performance, likely because raw loss trajectories are noisy and hard for the LLM to interpret. Adding **loss curves from the previous feedback round** further boosts performance by revealing the effectiveness of the last strategy and signs of under/overfitting. **Model parameters** also help to stabilize performance, often prompting the LLM to propose feature selection. Finally, **best validation loss so far** provides a complementary historical context, though its benefit is modest compared to other signals.

## 4.2 OVERALL EXPERIMENTS

In this section, our goal is to demonstrate the practical utility of the proposed method by evaluating its impact on: 1) optimizing final model performance, as reflected by identifying preprocessing strategies that achieve lower loss or higher accuracy, and 2) improving sampling efficiency, as reflected by discovering effective preprocessing strategies and achieving lower generalization error within fewer training epochs. These two aspects are essential for assessing the effectiveness of a preprocessing strategy search method: a useful search approach should be able to find preprocessing strategies that generalize well and at the same time discover them within fewer conversation rounds, reducing the overall training cost.

To ensure leakage safety, each of the six datasets is split into training and validation sets before any descriptive statistics are gathered. In the initial prompt, only the training data are used to compute the statistics for all features. No preprocessing is applied beforehand, preserving the primitiveness of the raw data. For fair comparison, we account for all training epochs consumed during the search for the best preprocessing method, since in practice this process is typically performed by humans through repeated cross-validation runs that require substantial computation. We call these epochs the searching epoch.

**Baselines.** We compare our method with four baselines: (1) **Zero-shot**, (2) **CoT (Chain-of-Thought**, (3) **Self-Refine**, with (i) **Single-Query Self-Refine**, and (ii) **Iterative Self-Refine**. Iterative Self-Refine can also be considered as an ablation study of the model performance feedback. These baselines are chosen to represent direct generation, reasoning-enhanced prompting, and iterative refinement. All methods are implemented under the same experimental settings to ensure fair comparison. For each dataset, we fix a set of random seeds and apply the same seeds across all methods to control variance. Unless otherwise specified, we use a batch size of 256. Since the optimal learning rate can vary depending on the preprocessing strategies, using a constant value for both baselines and our method maintains consistency and ensures fair comparison. Specifically, we set the learning rate to 0.01 for regression tasks and 0.001 for classification tasks. We apply early stopping with a patience of 10 epochs. Experiments on all datasets are implemented using the open-source LLM Deepseek-r1:32b, except for the Ames Housing dataset, for which we use Deepseek-r1:70b due to its more complex feature space (DeepSeek-AI et al., 2025).

**Comparison on Final Results**. We report task-specific evaluation metrics: accuracy(ACC) for classification datasets (ADU, OBS, HIG) and loss (RMSE/RMSLE as specified per task) for regression datasets (WQW, CAH, AMH). Table 3 summarizes the mean best score over multiple random seeds for each dataset. Compared to the baselines, our proposed method attains the highest accuracy

Table 3: Mean best loss/accuracy with standard error on 6 datasets. In the parentheses next to the dataset abbreviations are their corresponding evaluation metric. For this table we include extreme in average computation.

| | **ADU**(ACC) | **OBS**(ACC) | **HIG**(ACC) | **WQW**(RMSE) | **CAH**(RMSE) | **AMH**(RMSLE) |
|---|---|---|---|---|---|---|
| **LLM-Presto** | **0.853**(±0.001) | **0.855**(±0.005) | **0.658**(±0.020) | **0.732**(±0.020) | **0.567**(±0.090) | **0.166**(±0.036) |
| **Zero-shot** | 0.826(±0.040) | 0.812(±0.048) | 0.626(±0.017) | 0.790(±0.064) | 0.705(±0.061) | 0.381(±0.292) |
| **CoT** | 0.812(±0.045) | 0.762(±0.058) | 0.643(±0.007) | 0.745(±0.013) | 0.726(±0.050) | 4.743(±7.187) |
| **Self-refine I** | 0.845(±0.002) | 0.832(±0.013) | 0.626(±0.025) | 0.746(±0.002) | 0.701(±0.059) | 0.316(±0.248) |
| **Self-refine II** | 0.836(±0.016) | 0.828(±0.017) | 0.603(±0.028) | 0.812(±0.056) | 0.667(±0.032) | 0.196(±0.098) |

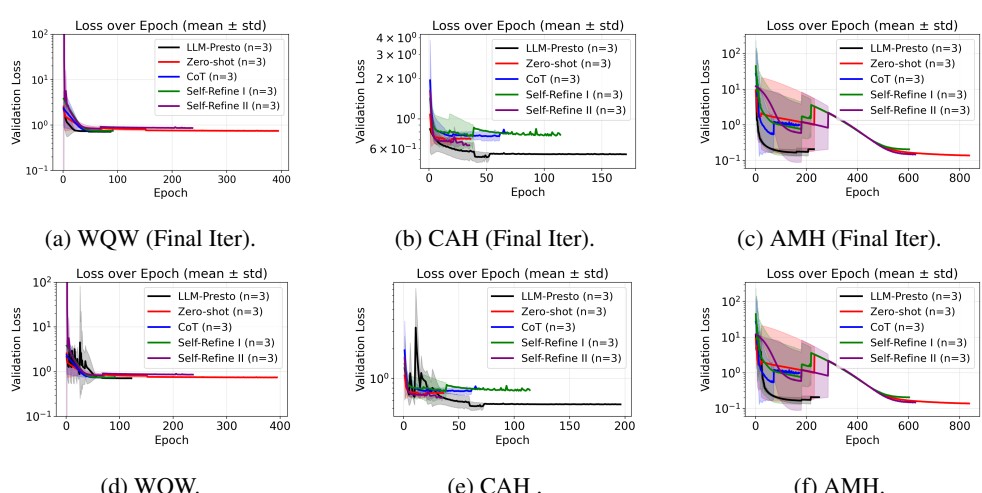

(a) WQW (Final Iter).     (b) CAH (Final Iter).     (c) AMH (Final Iter).

(d) WQW.     (e) CAH .     (f) AMH.

Figure 1: Validation loss curves. Average validation loss (mean ± std) versus epochs across datasets. The top row presents the final model training curves of LLM-Presto while the bottom row includes the losses during the searching epoch, which are the losses computed for preprocessing plan proposed the LLM when searching for an optimal one. The y-axis is in log scale, and number of runs averaged are shown in the parentheses.

in classification tasks and the lowest loss in regression tasks. These gains are obtained under the reduced-epoch feedback strategy established above, thus aligning the improvements in final metrics with our efficiency objectives.

We observe that the variance across seeds and datasets is not uniform, reflecting the distinct characteristics of datasets. For relatively simple datasets such as Wine Quality, which are already well-cleaned, the optimal preprocessing strategy can be identified in the very first round. More complex datasets require iterative refinement, and single-answer baseline methods frequently fail to find the best preprocessing strategy in one attempt, while Self-refine II often drifts away from promising initial solutions due to the lack of informative feedback. This limitation results in higher variance and degraded final performance.

Figures 1a 1b 1c and 2a 2b 2c provide detailed insight into convergence behavior and variance across datasets. In regression tasks, LLM-Presto achieves lower validation losses with smaller standard deviations and flatter curves in later epochs, indicating more stable convergence. In classification tasks, our approach also reaches higher final accuracy. The clear leftward shift of the curves demonstrates faster convergence and improved overall performance, suggesting that the preprocessing strategy identified are well optimized.

**Comparison on Sampling Efficiency**. To quantify sampling efficiency, we measure the number of training epochs required by each method to reach a certain target performance. The target is defined as the worst final evaluation metric achieved by the baselines, ensuring that each method has a valid reference point for comparison. Table 4 shows both the total number of epochs including the searching stage with $k = 5$ outside the parentheses and the number of epochs required to fully train

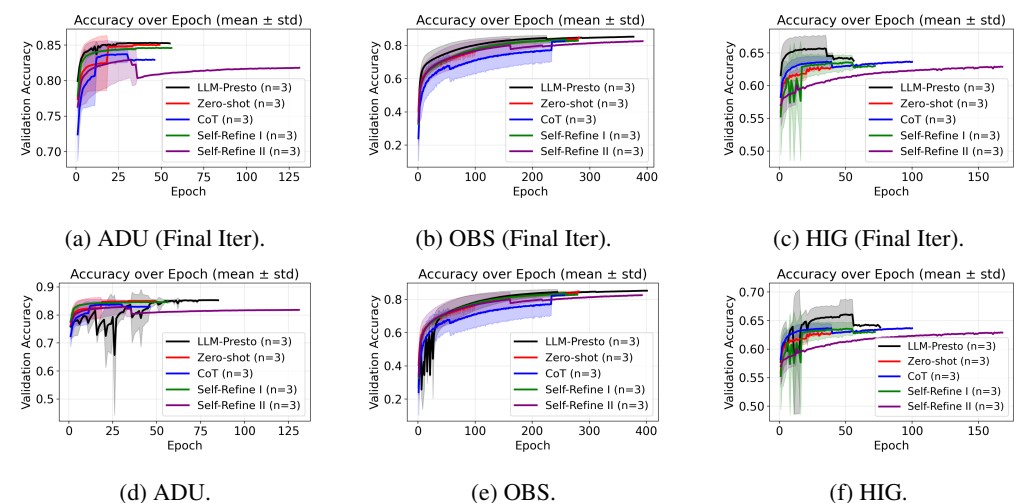

Figure 2: Validation accuracy curves. Average validation accuracy (mean ± std) versus epochs across datasets. The top row presents the final model training curves of LLM-Presto while the bottom row includes the accuracy of searching epoch. The number of runs averaged are shown in the parentheses.

Table 4: The epoch number used by each method to reach the target performance. The target chosen here is the worst final accuracy/loss that is achieved by the baselines other than extreme outliers, to ensure there is a valid number for each run. In the bracket is the average number of epochs to reach the target in the final model training stage.

|  | ADU(ACC) | OBS(ACC) | HIG(ACC) | WQW(RMSE) | CAH(RMSE) | AMH(RMSLE) |
|---|---|---|---|---|---|---|
| **Target** | **0.8182** | **0.7236** | **0.5715** | **0.8627** | **0.7839** | **0.9584** |
| **LLM-Presto** | **15.7(2.3)** | **61(37.7)** | **2.7(1)** | **8.7(8.7)** | **9(3.7)** | **22.3(5.7)** |
| **Zero-shot** | >35 | 57.7 | 2 | 79.7 | 4 | 129 |
| **CoT** | >38 | 114.7 | 1.3 | 24.3 | 26.7 | >390 |
| **Self-refine I** | 4 | 56.3 | 3 | 38.3 | 23.7 | 151 |
| **Self-refine II** | 46 | 65 | 5 | 97.3 | 4 | 180.3 |

the final model with the identified preprocessing strategy inside the parentheses. The two numbers are the same for the baselines.

As shown in Table 4, LLM-Presto consistently requires fewer epochs in the final training stage to achieve the same target performance compared to all baselines. For instance, on Ames Housing, our method reaches the target accuracy within 23 epochs when including the searching stage, and within only 6 epochs once the optimal preprocessing strategy is fixed. This is significantly lower than the other baselines. On simpler datasets, the final training stage of LLM-Presto always requires fewer epochs than the other baselines, and although the total epoch number can be slightly higher, it remains competitive by enabling the discovery of a more effective preprocessing strategy.

In addition to reducing the number of epochs to target, LLM-Presto also exhibits higher stability across different datasets. With $k = 5$, the iterative feedback method consistently achieves the lowest epoch numbers on all tasks, while the baselines' ranks vary across datasets. The robustness of our method helps to prove that the feedback pipeline generalizes well across diverse data characteristics and problem types, thus highlighting its reliability in practical settings.

In Table 5, we further illustrate sampling efficiency using the epoch number to the best evaluation metric. Here, the target performance is chosen as the mean best loss/accuracy achieved by our method. The results show that LLM-Presto reaches the target metric with substantially fewer epochs across datasets. In contrast, alternative prompting strategies require many more epochs, and often fail to reach the target even after the maximum training epoch limit. These observations demonstrate

Table 5: The epoch number used by each method to reach the target performance. The target set here is the min best accuracy/loss that is achieved by LLM-Presto. If the target is not achieved, we include the best metric up to the seen epochs in the parentheses. An extended version of this table with mean, min, and max listed is included in Appendix A.2 as Table 6 and 7

| Dataset | ADU(ACC) | OBS(ACC) | HIG(ACC) | WQW(RMSE) | CAH(RMSE) | AMH(RMSLE) |
|---|---|---|---|---|---|---|
| **Target** | 0.851 | 0.850 | 0.643 | 0.753 | 0.666 | 0.207 |
| LLM-Presto (final iter) | 17.3 | 239 | 19 | 14 | 15 | 110 |
| LLM-Presto (with searching) | 52.3 | 262 | 30 | 19 | 23 | 126 |
| Zero-shot | >84.3 | >569 | >100 | >100 | >67 | >843 |
| CoT | >100 | >805 | >50 | >50 | >100 | >1000 |
| Self-refine I | >100 | 387 | >53 | 54 | >72 | >577 |
| Self-refine II | >100 | >551 | >100 | >84 | >42 | >536 |

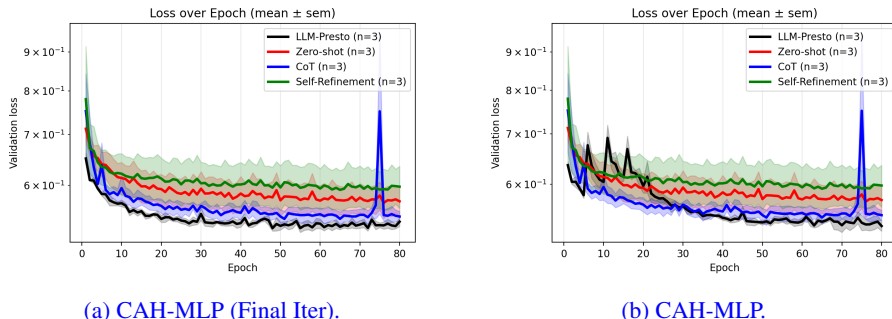

(a) CAH-MLP (Final Iter).          (b) CAH-MLP.

Figure 3: Validation loss curves over the first 80 epochs. Average validation loss (mean ± sem) versus epochs across datasets. The left figure presents the final model training curves of LLM-Presto while the right figure includes the loss during searching epoch. The number of runs averaged are shown in the parentheses.

that our method not only improves the final performance, but also achieves the desired performance or loss level more efficiently, thus reducing training cost.

As shown in Figure 1d 1e 1f and 2d 2e 2f, the curves for LLM-Presto increase higher for classification tasks and drop lower for regression tasks during later epochs, despite the oscillation in the early stages. Although these fluctuations, introduced by the iterative feedback step, reduce stability compared to single-shot baselines, the overall performance remains competitive. Even with oscillations, once a more effective preprocessing strategy is identified, the validation curve rapidly improves and can surpass the baseline curves during searching epochs. This pattern illustrates that the transient instability is offset by the capacity to discover stronger preprocessing strategies, eventually leading to better sampling efficiency.

In addition to linear models, to further evaluate the framework, we conduct experiments on a non-linear model using the California Housing dataset. The model is implemented as a multilayer perceptron (MLP) consisting of three hidden layers $[64, 64, 32]$ with ReLU activations. The batch size is 128 and the learning rate is 0.001. As shown in Figure 3, LLM-Presto (black curve) consistently outperforms the baselines. It is also more stable across different runs, as reflected by the smallest standard error as training progresses among all methods. These results demonstrate that the DNN model benefits greatly from the preprocessing strategies proposed by our framework. Thus, the findings indicate that LLM-Presto is effective for both linear and non-linear models that rely on data preprocessing.

## 5 CONCLUSION

This paper studies a simple yet effective iterative prompting method for optimizing column-wise data preprocessing strategies for linear models. We systematically investigated the types of information that should be provided to the LLM in order to generate preprocessing suggestions, and how these suggestions can be refined through iterative adjustments. Our approach leverages an efficient early-stopping criterion—evaluating validation error within a fixed training budget—to select the best preprocessing strategy. Through extensive ablation studies, we verified the importance of each design choice, including the type of information fed to the LLM and the role of early stopping in evaluation. Overall, our method is capable of identifying preprocessing strategies that yield strongest generalization performance, while also demonstrating significantly improved sample efficiency compared to a variety of baselines—even after accounting for the training episodes consumed during the preprocessing search.

**Limitations and future work**. We highlight three main limitations. First, the performance of LLMs may degrade when the number of features is large, reflecting the broader challenge of handling long-context inputs (Liu et al., 2024); in such cases, the suggestions may become less specific. Second, our work focuses primarily on linear models, motivated by their simplicity and interpretability in line with prior literature (Hastie et al., 2009). Extending the approach to deep learning settings would be a natural and valuable direction for future research. Third, we have not yet explored integrating our method into existing AutoML frameworks (Feurer et al., 2015; Olson et al., 2016), which could further enhance its practical utility. While such integration would require significant engineering effort, we believe it is an important avenue for future work.

ETHICS STATEMENT

This work aims to automate data preprocessing to improve the efficiency of machine learning work-flows. We acknowledge the broader ethical context of this research. While our intention is to reduce the significant time and cost of manual pipeline design for legitimate applications, we recognize that any efficiency tool has the potential for dual use. The primary ethical consideration is that the preprocessing strategies generated by our LLM-based framework could, if applied without scrutiny, perpetuate or amplify biases present in the underlying data or the LLM itself, leading to unfair models.

To address this, we emphasize that our method is designed as an assistive tool for practitioners, not a fully autonomous system. Responsible use requires validating suggested preprocessing steps against fairness and domain-specific criteria. Our empirical evaluation used standard, publicly available benchmarks in compliance with their licenses.

REPRODUCIBILITY STATEMENT

We have taken several steps to ensure the reproducibility of our work. The main paper details the proposed algorithm and its design choices, while the appendix documents implementation specifics, including parameter sweeps, random seeds, software libraries, and computing environment. All datasets used in our experiments are publicly available, with data sources clearly provided. A complete code repository with scripts to reproduce all experiments will be released upon publication.

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

## A   APPENDIX

This appendix introduces additional background about mathematical insights into why preprocessing matters, additional empirical results that may be of interest to readers, and experimental details that used to reproduce our experiments.

### A.1   MORE MATH DETAILS

**Convergence perspective.** Consider the empirical square-loss, along with the gradient and Hessian notations:

$$f(w) = \frac{1}{2n}\|Xw - y\|_2^2, \qquad \nabla f(w) = \frac{1}{n}X^\top(Xw - y), \qquad H := \nabla^2 f(w) = \frac{1}{n}X^\top X.$$

Consider gradient descent with step size $\eta > 0$. It follows

$$w_{t+1} = w_t - \eta \nabla f(w_t), \quad e_{t+1} = (I - \eta H)e_t, \quad e_t := w_t - w^\star.$$

If $0 < \eta < 2/\lambda_{\max}(H)$, then $\|e_t\|_2 \le \rho^t \|e_0\|_2$ with $\rho = \max_i |1 - \eta\lambda_i(H)|$. Using the optimal fixed step $\eta^\star = \frac{2}{\lambda_{\max}(H) + \lambda_{\min}(H)}$ yields the linear rate

$$\|e_t\|_2 \le \left(\frac{\kappa - 1}{\kappa + 1}\right)^t \|e_0\|_2, \qquad \kappa := \frac{\lambda_{\max}(H)}{\lambda_{\min}(H)}.$$

Thus, convergence speed is controlled by the condition number $\kappa$.

Here are some examples of how preprocessing might improve condition number. Applying an invertible feature transform $Z = XP$ replaces $H$ by $P^\top HP$ (Gutman & Peña, 2021). Good preprocessing chooses $P$ so that $P^\top HP$ is closer to $I$ (smaller $\kappa$).

Scaling/standardization (e.g., z-scores) approximately equalizes column norms and typically reduces $\kappa$. In the toy case of uncorrelated columns with variances $\sigma_j^2$, $H = \mathrm{diag}(\sigma_j^2)$ so $\kappa = \max_j \sigma_j^2 / \min_j \sigma_j^2$; standardizing ($\sigma_j = 1$) gives $\kappa = 1$.

With centered $X = U\Sigma V^\top$, set $Z := XV\Sigma^{-1}\sqrt{n}$ so that $\frac{1}{n}Z^\top Z = I$. Then $H = I$ and gradient descent with $\eta = 1$ converges in a single step in the transformed coordinates. Centering $X$ and $y$ (when fitting an intercept) decouples the intercept and generally improves conditioning.

**Generalization perspective**. If the preprocessing is invertible and linear (e.g., scaling by $S$) and the same transform is applied to train and test features, then OLS predictions are invariant: using $Z = XS$ merely reparameterizes $w$. Generalization changes arise when (i) regularization or early stopping is used, or (ii) the transform is non-invertible (e.g., PCA truncation, winsorization, binning).

Consider ridge on $Z = XS$:

$$\hat{\beta} = \arg \min_\beta \frac{1}{2n}\|Z\beta - y\|_2^2 + \frac{\lambda}{2}\|\beta\|_2^2.$$

In original coordinates $\theta = S\beta$,

$$\min_\theta \ \frac{1}{2n}\|X\theta - y\|_2^2 + \frac{\lambda}{2}\theta^\top S^{-2}\theta.$$

Thus, without standardization, ridge (and similarly Lasso) imposes uneven feature-wise penalties (small-variance features are penalized more), which can increase bias and harm test error (Hastie et al., 2009; Hoerl & Kennard, 1970). Standardization makes the penalty more isotropic and typically improves generalization.

One might also consider the perspective of early stopping as spectral shrinkage (Ali et al., 2019; Sonthalia et al., 2024). After $t$ GD steps,

$$\hat{w}_t = g_t(H)\frac{1}{n}X^\top y, \qquad g_t(\lambda) = \frac{1 - (1 - \eta\lambda)^t}{\lambda},$$

so each eigendirection of $H$ is shrunk by $g_t(\lambda)$. A spread spectrum induces anisotropic shrinkage (overfitting high-variance directions, underfitting low-variance ones). Preprocessing that flattens the spectrum (scaling, whitening, PCA) makes this implicit regularization more uniform, often reducing test error.

## A.2 ADDITIONAL RESULTS

This subsection provides details and further analysis of the empirical study in Section 4.

**Comparison on Feedback Budget** We compare how the searching epoch number $k$ in the feedback affects the efficiency of finding the optimal preprocessing strategy. We also test whether $k$ is stable regarding different learning rates. By optimal we mean the optimal of that specific run. As

shown in Table 1, setting the feedback epoch number to 5 achieves the best balance between optimization quality and computational efficiency. With LR = 0.005, it reaches the lowest loss (0.5970) while requiring substantially fewer epochs than larger settings. Similar trends hold for LR = 0.01 and 0.1, where $k = 5$ matches or outperforms $k = 10$ in loss (0.5669 vs. 0.5605 at LR = 0.01; 0.5767 vs 0.5772 at LR = 0.1), and with around $50\%$ fewer epochs (67 vs. 124 and 54 vs. 126). In the contrary, $k = 1$ is unstable, often leading to higher losses and failing to provide reliable feedback for strategy selection. On the other hand, increasing $k \geq 10$ does not significantly improves the performance compared to $k$ of 5, and because of the higher costs of training it leads to diminishing returns. $k = 5$ is also insensitive to the change in learning rate, as observed from its consistent good performance regarding all three learning rates. In conclusion, $k = 5$ is both efficient and stable in finding the optimal preprocessing strategy. Therefore, we choose $k = 5$ as the default.

**Effect of Large Feedback Budget**   As shown in Table 1, we observe that the $k = 20$ experiments occasionally exhibit degraded performance compared to smaller feedback budgets. A plausible explanation is that, since the model is often trained close to convergence within the $k = 20$ budget, the LLM infers that the current preprocessing strategy is already sufficiently effective. Consequently, the LLM tends to shift its focus toward feature selection or dimensionality reduction, rather than exploring additional preprocessing strategies. For example, it frequently proposes the use of PCA to compress the feature space, even when the number of features is already small.

```
**Dimensionality Reduction (PCA)**
- **Objective:** Reduce the number of features while retaining
most of the variance.
- **Implementation:**
```python
from sklearn.decomposition import PCA
# Apply PCA to reduce dimensionality
pca = PCA(n_components=0.95)
# Retain 0.95 of variance
principal_components = pca.fit_transform(
    data[['feature1', 'feature2', ...]]
)
# Replace the original features with PCA components
data_pca = pd.DataFrame(
    principal_components,
    columns=['PC1', 'PC2', ...]
)
```

While dimensionality reduction and feature selection is expected with the LLM's objective of optimizing efficiency, it leads to a worse performance. This is because the model prioritizes improving efficiency over refining the preprocessing pipeline as a whole, thus discarding potentially informative features.

We further compare the effect of restricting the LLM from using PCA at different values of $k$. For $k = 20$, explicitly forbidden PCA indeed prevents the LLM from prematurely proposing PCA-based preprocessing. However, as shown in Figure 4, the overall performance is still poorer compared to the $k = 5$ setting. There are two key reasons. First, when $k = 20$, the validation loss has already largely converged during the search phase, causing the LLM to believe that the current preprocessing strategy is already near optimal. As a result, it stops generating new preprocessing suggestions. This never occurs when $k = 5$. Second, the large $k$ prolongs the search process. $k = 20$ produces fewer meaningful updates and wastes computational budget, making it a less efficient choice overall.

**Additional Analysis of Epochs to Reach Target**   Table 6 and 7 report the detailed number of training epochs required by each method to achieve the target performance.

LLM-Presto consistently reach the target in significantly fewer epochs compared to the baselines whether or not the searching epoch number is taken into account. Even when additional searching increases the total number of epochs, the process remains competitive and is capable of discover stronger preprocessing strategies. In contrast, the baselines often fail to meet the target. They

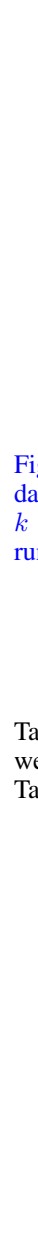
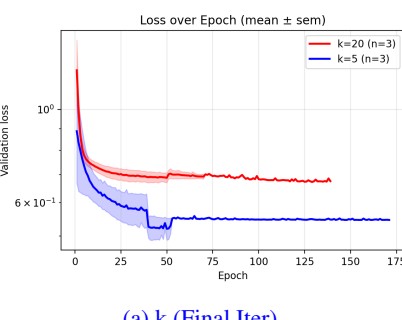
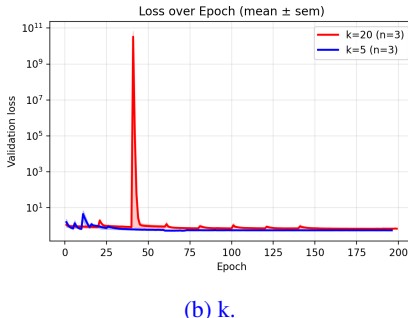

(a) k (Final Iter).          (b) k.

Figure 4: Validation loss curves. Average validation loss (mean $\pm$ sem) versus epochs across datasets. The left figure presents the final model training curves of LLM-Presto with $k = 5$ and $k = 20$ respectively, and the right figure includes the loss during searching epoch. The number of runs averaged are shown in the parentheses.

Table 6: Epochs to reach the target accuracy for ADU, OBS, and HIG. If the goal is not achieved, we include the best metric within the seen epochs in parentheses. This table is the full version of Table 5 from Section 4.2 on classification tasks.

(a) ADU

|  | Mean | Max | Min |
|---|---|---|---|
| **Target** | | **0.851** | |
| LLM-Presto (final iter) | 17.3 | 20 | 16 |
| LLM-Presto (with searching) | 52.3 | 66 | 50 |
| Zero-shot | $> 84.3$ | $> 100(0.780)$ | 53 |
| CoT | $> 100$ | $> 100(0.760)$ | $> 100(0.844)$ |
| Self-refine I | $> 100$ | $> 100(0.843)$ | $> 100(0.847)$ |
| Self-refine II | $> 100$ | $> 100(0.817)$ | $> 100(0.847)$ |

(b) OBS

|  | Mean | Max | Min |
|---|---|---|---|
| **Target** | | **0.850** | |
| LLM-Presto (final iter) | 239 | 332 | 110 |
| LLM-Presto (with searching) | 262 | 357 | 130 |
| Zero-shot | $> 569$ | $> 1000(0.849)$ | 291 |
| CoT | $> 805$ | $> 1000(0.803)$ | 415 |
| Self-refine I | 387 | 475 | 267 |
| Self-refine II | $> 551$ | $> 1000(0.844)$ | 166 |

(c) HIG

| Epochs | Mean | Max | Min |
|---|---|---|---|
| **Target** | | **0.643** | |
| LLM-Presto (final iter) | 19 | 45 | 1 |
| LLM-Presto (with searching) | 30 | 65 | 9 |
| Zero-shot | $> 100$ | $> 100(0.607)$ | $> 100(0.637)$ |
| CoT | $> 50$ | $> 100(0.637)$ | 20 |
| Self-refine I | $> 53$ | $> 100(0.637)$ | 20 |
| Self-refine II | $> 100$ | $> 100(0.573)$ | $> 100(0.623)$ |

Table 7: Epochs to reach the target loss for WQW, CAH, AMH. If the goal is not achieved, we include the best metric within the seen epochs in parentheses. This table is the full version of Table 5 from Section 4.2 on regression tasks.

(a) WQW

| Epochs | Mean | Max | Min |
|---|---|---|---|
| Target | 0.753 | | |
| LLM-Presto (final iter) | 14 | 25 | 2 |
| LLM-Presto (with searching) | 19 | 35 | 2 |
| Zero-shot | > 100 | > 100(0.867) | > 100(0.767) |
| CoT | > 50 | > 100(0.757) | 5 |
| Self-refine I | 54 | 62 | 50 |
| Self-refine II | > 84 | > 100(0.893) | 51 |

(b) CAH

| Epochs | Mean | Max | Min |
|---|---|---|---|
| Target | 0.666 | | |
| LLM-Presto (final iter) | 15 | 29 | 1 |
| LLM-Presto (with searching) | 23 | 43 | 4 |
| Zero-shot | > 67 | > 100(0.747) | 2 |
| CoT | > 100 | > 100(0.784) | > 100(0.691) |
| Self-refine I | > 72 | > 100(0.754) | 16 |
| Self-refine II | > 42 | > 100(0.700) | 6 |

(c) AMH

| Epochs | Mean | Max | Min |
|---|---|---|---|
| Target | 0.207 | | |
| LLM-Presto (final iter) | 110 | 208 | 59 |
| LLM-Presto (with searching) | 126 | 228 | 69 |
| Zero-shot | > 843 | > 1000(0.706) | 529 |
| CoT | > 1000 | > 1000(8.303) | > 1000(0.211) |
| Self-refine I | > 577 | > 1000(0.461) | 156 |
| Self-refine II | > 536 | > 1000(0.274) | 94 |

converge around the best observed metric shown in the parentheses. This indicates that without iterative feedback refinement, these approaches may stall below the desired performance.

Data complexity influences both the absolute epoch number and the relative gap between methods. For example, on simpler datasets such as WQW, LLM-Presto reaches the target in fewer than 20 epochs, while some baselines barely reach the target within the total epoch number limit. On more challenging datasets such as ADU and AMH, the gap enlarges, with LLM-Presto still reaching the target while other methods fail.

## A.3 EXPERIMENT DETAILS FOR REPRODUCIBLE RESEARCH

**Datasets.** In Table 8 we present the full description of the datasets we use. All datasets are obtained from public repositories: Adult Census Income (Becker & Kohavi, 1996), Obesity Risk (Reade & Chow, 2024), Higgs Boson (Baldi et al., 2014; Whiteson, 2014)(UCI dataset version 1), Wine Quality (White) (Cortez et al., 2009a;b), California Housing (Kelley Pace & Barry, 1997)( loaded via sklearn.datasets.fetch_california_housing ), and Ames Housing (Cock, 2011).

**Random Seed.** For reproducibility, we fixed three random seeds: 1757860097, 1758570208, and 1758568404. For each dataset, we evaluated all methods under the same three random seeds to ensure fair comparison under identical initialization and data shuffling conditions.

**Software libraries.** Experiments were implemented in Python 3.10.12 and rely on a standard ML stack, including PyTorch 2.7.1, scikit-learn 1.7.0, NumPy 2.2.6, and pandas 2.3.1.

Table 8: Description of selected datasets. Num stands for numerical and Cat stands for categorical. The first three datasets are used for Classification task, the last three are used for Regression task. For classification tasks, we choose accuracy as the evaluation metric. For regression tasks, we use root mean square error (RMSE) for Wine Quality and California Housing dataset, and root mean square logarithmic error (RMSLE) for the Ames Housing, as this is the evaluation metric used by the Kaggle Housing Prices Competition (DanB, 2018).

| Dataset Name | Resource | Feature Type | Feature Number | Sample number |
|---|---|---|---|---|
| Adult Census Income | UCI | Num & Cat | 14 | 48842 |
| Obesity Risk | Kaggle | Num & Cat | 17 | 20758 |
| Higgs Boson | UCI | Num | 28 | 98050 |
| Wine Quality (White) | UCI | Num | 11 | 4898 |
| California Housing | scikit-learn | Num | 8 | 20640 |
| Ames Housing | OpenML | Num & Cat | 79 | 1460 |

**Computing environment.** All experiments were executed on a Linux workstation Linux-5.15.0-131-generic-x86 64-with-glibc2.35. GPU runs used an NVIDIA RTX A5000 (24 GB) with Driver 560.35.05 and CUDA 12.6.

**Parameter sweeps.** We varied learning rates and used a fixed batch size. We swept the learning rate over 0.001, 0.01 with batch size set to 256 for all runs. Following early tuning, we used learning rate 0.01 for regression tasks on Ames Housing, California Housing, and Wine Quality, and 0.001 for classification tasks on Adult Census, Higgs, and Obesity Risk.

**Data split.** For all experiments, the train-validation split is set to 80:20. The same split was applied across all methods to ensure fair comparison.

**Model Description Summary.** Table 9 summarizes the downstream model setup.

Table 9: Model configurations for linear models and MLP.

| Model Type | Linear regression model | Linear classification model (binary & multiclass) | Multilayer Perceptron (MLP), feedforward fully-connected neural network |
|---|---|---|---|
| **Architecture** | N/A | N/A | Hidden layers: [64, 64, 32] Activation: ReLU |
| **Loss function** | Mean Squared Error (MSE) | BCEWithLogits (binary) / CrossEntropy (multiclass) | Mean Squared Error (MSE) |
| **Optimizer** | Adam | Adam | Adam |
| **Learning rate** | Specified per task | Specified per task | Specified per task |
| **Batch size** | Specified per task | Specified per task | Specified per task |
| **Early stopping** | Patience = 10 | Patience = 10 | Patience = 10 |
| **Weight decay** | 0.0 | 0.0 | 0.0 |
| **Initialization** | PyTorch default initialization | PyTorch default initialization | PyTorch default initialization |
| **Train/val split** | 80/20 | 80/20 | 80/20 |

**Prompt and response example.** The structure and content of the initial prompts, the feedback prompts, and an example of the LLM's response are provided below. For nonlinear models such as MLPs, feedback prompt contents related to weights and biases are omitted, as such information is meaningless for this type of architecture.

**Initial Prompt**

```
You are a senior ML engineer. You are assisting with data
    preprocessing used before training a {model_desc['type']} model.

DATASET
- Number of samples: {stats['n_instances']}
- Number of features: {stats['n_features']}
- Feature names: {stats['feature_names']}
- Target(y): {target_description}

SAMPLE DATA (random {n_samples} rows):
{preview}

SUMMARY STATS (training set)
Numerical Features:
{num_summary}

Categorical Features:
{cat_summary}

TARGET MODEL CONTEXT (for awareness only; do NOT change or tune):
- Model: {type}
- Task: Regression
- Number of Parameters: {parameters}
- Loss Function: {loss_fun}
- Optimizer: {optimizer}
- Learning Rate: {learning_rate}
- Batch Size: {batch_size}
- Target definition: {target_description}

TASK
Propose a concrete preprocessing pipeline for the already-split
    dataset (fit on TRAIN only; apply to VAL with the exact fitted
    parameters).
Include data cleaning, missing value handling, encoding, scaling,
    outlier clipping, and feature engineering decisions, etc.
Do NOT suggest or modify anything related to training (no epochs/
    optimizers/regularization/architecture/hyperparameters).

RESTRICTIONS
- Be fully prescriptive: no options, no vague language, no "if/then
    " left to me.
- State exactly which features each step applies to.
- Provide a strict total order of steps, with deterministic
    thresholds and parameters.
- All steps must be leakage_safe = true.

OUTPUT SCHEMA (return exactly in this structure):
1) "per_feature_decisions": a list where each item is:
    - "feature": feature name (string)
    - "dtype": detected type ("numeric", "categorical", "boolean", "
        datetime", "text", "id_like", "drop")
    - "stats": dictionary with relevant dataset-specific stats (
        missing_rate, n_unique, skewness if numeric, etc.)
    - "preprocessing": ordered list of actions applied ONLY to this
        feature (e.g. ["impute: median", "clip: p0.5-p99.5", "scale:
        StandardScaler"])
2) "global_steps": ordered list of dataset-wide preprocessing
    actions, where each item is:
    - "name": short snake_case
```

```
    - "applies_to": list of feature names or special tokens ["
        ALL_NUMERIC"], ["ALL_CATEGORICAL"], ["TARGET"]
    - "action": imperative description with deterministic thresholds
        /parameters
    - "leakage_safe": true
3) "pseudocode": concise end-to-end pseudocode showing how
    preprocessing is fit on TRAIN and then applied to VAL
4) "python": a single deterministic Python function.
```

### Feedback Prompt

```
After applying the preprocessing you suggested and training (the
    losses are inverse transformed):
Epochs trained: {training_log['epochs_trained']}
Train loss: {train_losses_str}
Validation loss: {val_losses_str}
Best loss achieved over the {iter} iterations so far: {best_loss}
Learned parameters of the model:
| Feature Name | Weight   |
|--------------|----------|
{table_rows}

**Bias:** {bias_str}

Please re-evaluate the current preprocessing strategy and propose a
    revised preprocessing plan based on the training results above.
Requirements:
- Follow the same specificity rules as before.
- Focus only on data preprocessing.
- State exactly which feature(s) each step applies to.
- Please explicitly state the sequence of how the steps should be
    implemented.
- Do not give vague instructions or optional steps.
- Do not leave any decisions to me. Steps must be actionable and
    immediately implementable.
- Please provide Python code or pseudo code in the end that
    performs your preprocessing strategy.
- If no changes are necessary, respond exactly with:
    "No further changes needed."
- I will continue asking you for more suggestions until you say "No
    further changes needed".
- Please be aware that changes are always encouraged, but must be
    reasonable and justified.
```

### Example Response from LLM (California Housing Dataset)

```json
{
    "steps": [
        {
            "name": "fill_missing_median",
            "applies_to": ["ALL_NUMERIC"],
            "action": "Replace all missing values with median of
                training set.",
            "leakage_safe": true
```

```
            },
            {
                "name": "log_transform_population",
                "applies_to": ["Population"],
                "action": "Apply log transformation: population = log(
                    population + 1)",
                "leakage_safe": true
            },
            {
                "name": "log_transform_ave_occupancy",
                "applies_to": ["AveOccup"],
                "action": "Apply log transformation: ave_occupancy =
                    log(ave_occupancy + 1)",
                "leakage_safe": true
            },
            {
                "name": "normalize_med_inc",
                "applies_to": ["MedInc"],
                "action": "Normalize to z-score using training set mean
                    and std.",
                "leakage_safe": true
            },
            {
                "name": "normalize_house_age",
                "applies_to": ["HouseAge"],
                "action": "Normalize to z-score using training set mean
                    and std.",
                "leakage_safe": true
            },
            {
                "name": "normalize_ave_rooms",
                "applies_to": ["AveRooms"],
                "action": "Normalize to z-score using training set mean
                    and std.",
                "leakage_safe": true
            },
            {
                "name": "normalize_ave_bedrms",
                "applies_to": ["AveBedrms"],
                "action": "Normalize to z-score using training set mean
                    and std.",
                "leakage_safe": true
            },
            {
                "name": "normalize_latitude",
                "applies_to": ["Latitude"],
                "action": "Normalize to z-score using training set mean
                    and std.",
                "leakage_safe": true
            },
            {
                "name": "normalize_longitude",
                "applies_to": ["Longitude"],
                "action": "Normalize to z-score using training set mean
                    and std.",
                "leakage_safe": true
            },
            {
                "name": "cap_outliers_95_percentile",
                "applies_to": ["ALL_NUMERIC"],
                "action": "Cap values at 95th percentile for each
                    feature based on training data.",
```

```
                    "leakage_safe": true
                }
            ],
            "pseudocode": [
                "// Fit preprocessors on training set",
                "medians = train_df.median()",
                "train_filled = train_df.fillna(medians)",
                "val_filled = val_df.fillna(medians)",
                "",
                "// Apply log transforms to Population and AveOccupancy",
                "train_filled['Population'] = np.log(train_filled['
                    Population'] + 1)",
                "val_filled['Population'] = np.log(val_filled['Population']
                     + 1)",
                "train_filled['AveOccup'] = np.log(train_filled['AveOccup']
                     + 1)",
                "val_filled['AveOccup'] = np.log(val_filled['AveOccup'] +
                    1)",
                "",
                "// Normalize numeric features using training statistics",
                "normalize_features = ['MedInc', 'HouseAge', 'AveRooms', '
                    AveBedrms', 'Latitude', 'Longitude']",
                "means = train_filled[normalize_features].mean()",
                "stds = train_filled[normalize_features].std()",
                "",
                "// Apply normalization to training and validation sets",
                "train_normalized = (train_filled[normalize_features] -
                    means) / stds",
                "val_normalized = (val_filled[normalize_features] - means)
                    / stds",
                "",
                "// Reattach non-normalized columns",
                "train_df_processed = train_filled.copy()",
                "val_df_processed = val_filled.copy()",
                "train_df_processed[normalize_features] = train_normalized
                    ",
                "val_df_processed[normalize_features] = val_normalized",
                "",
                "// Cap outliers at 95th percentile for each feature",
                "cap_percentile = 0.95",
                "caps = train_df_processed.quantile(cap_percentile)",
                "train_df_processed = train_df_processed.where(
                    train_df_processed <= caps, caps, axis=1)",
                "val_df_processed = val_df_processed.where(val_df_processed
                     <= caps, caps, axis=1)"
            ],
            "python": [
                "def preprocess(train_df, val_df):",
                "    # Step 1: Fill missing values with median (leakage-
                    safe)",
                "    medians = train_df.median()",
                "    train_filled = train_df.fillna(medians)",
                "    val_filled = val_df.fillna(medians)",
                "",
                "    # Step 2 & 3: Log transform Population and
                    AveOccupancy",
                "    train_filled['Population'] = np.log(train_filled['
                    Population'] + 1)",
                "    val_filled['Population'] = np.log(val_filled['
                    Population'] + 1)",
                "    train_filled['AveOccup'] = np.log(train_filled['
                    AveOccup'] + 1)",
```

```
            "     val_filled['AveOccup'] = np.log(val_filled['AveOccup']
                + 1)",
            "",
            "    # Step 4-9: Normalize selected features",
            "     normalize_features = ['MedInc', 'HouseAge', 'AveRooms
                ', 'AveBedrms', 'Latitude', 'Longitude']",
            "     means = train_filled[normalize_features].mean()",
            "     stds = train_filled[normalize_features].std()",
            "",
            "    # Apply normalization to training and validation sets
                ",
            "     train_normalized = (train_filled[normalize_features] -
                means) / stds",
            "     val_normalized = (val_filled[normalize_features] -
                means) / stds",
            "",
            "    # Reattach non-normalized columns",
            "     train_df_processed = train_filled.copy()",
            "     val_df_processed = val_filled.copy()",
            "     train_df_processed[normalize_features] =
                train_normalized",
            "     val_df_processed[normalize_features] = val_normalized
                ",
            "",
            "    # Step 10: Cap outliers at 95th percentile (leakage-
                safe)",
            "     cap_percentile = 0.95",
            "     caps = train_df_processed.quantile(cap_percentile)",
            "     train_df_processed = train_df_processed.where(
                train_df_processed <= caps, caps, axis=1)",
            "     val_df_processed = val_df_processed.where(
                val_df_processed <= caps, caps, axis=1)",
            "",
            "    # Collect fitted parameters",
            "     fitted_params_dict = {",
            "         'medians': medians.to_dict(),",
            "         'means': means.to_dict(),",
            "         'stds': stds.to_dict(),",
            "         'caps': caps.to_dict()",
            "     }",
            "",
            "     return train_df_processed, val_df_processed,
                fitted_params_dict"
    ]
  }
  ```
```

### A.4 LLM USAGE

Large Language Models (LLMs) were used in the preparation of this paper for limited editorial assistance: grammar checking, phrasing refinement, and improvement of clarity. They were also used to help with LaTeX table formatting and to support, but not replace, parts of the literature search. All research ideas, methodological design, experiments, analyses, and interpretations are solely the work of the authors.

