# OpenReview forum: "AN ITERATIVE PROMPTING FRAMEWORK FOR LLM-BASED DATA PREPROCESSING"
_ICLR.cc/2026/Conference — Submitted to ICLR 2026_

### Official Review · Reviewer_EXXJ · 2025-10-26

**Soundness:** 3
**Presentation:** 2
**Contribution:** 2
**Rating:** 2
**Confidence:** 4

**Summary:**

This paper proposes LLM-Presto, an iterative prompting framework that automates data preprocessing with large language models. The method iteratively generates and refines preprocessing strategies based on structured feedback from validation results under a fixed training-epoch budget. It grounds the design in theoretical analysis of convergence and generalization for linear models and evaluates performance on six tabular datasets, showing improved efficiency and comparable or better accuracy than several prompting-based baselines.

**Strengths:**

- The paper is well-motivated and theoretically grounded, offering a clear connection between data preprocessing, conditioning, and generalization behavior, which gives the approach conceptual depth beyond empirical heuristics.
- The experimental evaluation is carefully structured, with ablation studies, baseline comparisons, and reproducibility details that enhance credibility and provide solid evidence for the efficiency gains claimed.

**Weaknesses:**

- The methodological novelty is limited, as the framework mostly combines existing iterative prompting and feedback techniques without introducing substantially new algorithmic ideas.
- The experiments are restricted to simple linear models and small-scale six tabular datasets, which constrains the generality and practical relevance of the conclusions.
- The analysis of performance variance and stability remains shallow, with oscillations noted but not rigorously explained or quantified.
- Finally, the paper’s presentation could be improved by adding clearer figures or concrete examples of LLM-generated preprocessing strategies to make the iterative process more intuitive and interpretable.

**Questions:**

See #Weakness

---

> ### Author Response · Authors · 2025-12-01
>
> Thank you for your thoughtful review and constructive feedback. We address the main issues below and will incorporate all minor suggestions in future versions.
>
> **1. Dataset scale and generality**
>
> Our datasets include sample numbers from 10k to 100k and feature numbers from 8 to 79. These sizes are representative of standard real-world tabular datasets. Per your instruction, we now include experiments with a multilayer perceptron to show that the framework extends beyond linear models in Section 4.2.
>
> **2. Performance oscillations and stability**
>
> Thank you for highlighting this point. The figures labeled with “final iter” show the training period curve after the preprocessing strategy is fixed, whereas figures without this label include both searching period and training period. Presenting these two types of figures is intentional. It allows a rigorous comparison between the final training behaviour and the search behaviour. The oscillations in the latter occur because the LLM alternates between competing preprocessing strategies during searching. Since each switch resets the model training from scratch, it is expected that the validation loss cannot monotonically decrease.
>
> Importantly, Figure 1 and 2 also show that even when including the oscillations during searching, our iterative method still consistently outperforms the baselines. This demonstrates that the oscillation behavior we observe is not instability, but rather an effective search process that converges toward better preprocessing choices.
>
> **3. Presentation and interpretability**
>
> We appreciate your suggestion. To improve interpretability, we have now included examples of LLM-generated transformations in the Appendix A.3. If the reviewer has preferred visualization styles, we welcome suggestions.

---

### Official Review · Reviewer_3cGo · 2025-10-31

**Soundness:** 2
**Presentation:** 2
**Contribution:** 2
**Rating:** 4
**Confidence:** 4

**Summary:**

This paper proposes LLM-Presto, an iterative prompting framework that uses LLMs to design and refine ML data preprocessing pipelines. LLM-Presto integrates LLM-generated preprocessing suggestions with few-epoch downstream validation feedback. Experiments show that LLM-Presto improves downstream performance with lower total training cost.

**Strengths:**

- **Significance**: The paper demonstrates a practical path toward integrating LLM reasoning into data-centric AutoML, showing potential for cost-efficient and interpretable pipeline design.
- **Clarity**: The framework is clearly structured and easy to follow.

**Weaknesses:**

1. **Limited methodological novelty**. The framework mainly combines existing elements, such as LLM prompting, few-epoch evaluation, and iterative refinement, without introducing a fundamentally new mechanism. The theoretical discussion on convergence and generalization is motivating but doesn’t serve as an integral part of the method.

2. **Insufficient experiment**. The experiments lack diversity on ML algorithms, so it’s difficult to judge whether the framework generalizes beyond specific settings. Also, the method has not been compared with traditional AutoML methods to show competitive performance.

3. **Presentation issues**. The term *iteration* is used inconsistently (sometimes referring to training epochs, other times to LLM feedback rounds), which causes confusion. The lack of explicit description of downstream models and implementation details, e.g. prompts,  also limits reproducibility.

**Questions:**

1. What exactly are the “losses during the search epoch” in Figure 1(d/e/f). Can you provide detailed explanations?
2. Efficiency is measured only by training cost. Have you taken LLM inference latency into consideration, since it may affect the practical runtime advantage?
3. How are the textual preprocessing suggestions from the LLM grounded into executable transformations?
4. Other questions are reflected in weaknesses.

---

> ### Author Response · Authors · 2025-12-01
>
> Thank you for your careful assessment of our work. We address your concerns below.
>
> **1. Model diversity and AutoML comparison**
>
> Regarding ML model diversity, we focus on linear models to isolate preprocessing effects and avoid confounding factors from model-specific behaviors. For your reference, new experiments with a multilayer perceptron as the downstream model are added at the end of Section 4.2. The results indicate the method’s effectiveness in both linear and nonlinear settings.
>
> Regarding the lack of AutoML comparisons, our goal in this work is to study LLM-generated preprocessing strategies, not the whole ML pipeline (which typically concerns more than preprocessing), and therefore we choose baselines that represent different principal prompting paradigms used in LLM-driven pipeline design. They represent the closest conceptual baselines to our method. We agree that AutoML is an important comparison class, but it addresses a different objective: AutoML optimizes model hyperparameters/selection  and searches full pipelines, while our work focuses only on the effect of preprocessing suggested by LLMs.
>
> In addition, many classic AutoML frameworks (e.g. Auto-weka\[1\], Auto-sklearn\[2\], TPOT\[3\]) assume clean tabular inputs. When dealing with raw data with heavy missing values, mixed datatypes, and unparsed date/time, these systems often require manual preprocessing. Our method outweighs them in data processing by allowing a fully automated process.
>
> That said, if the reviewer believes that a specific AutoML system is necessary to include, we would be happy to add it and kindly request clarification on which system they consider most appropriate.
>
> **2. Presentation issues**
>
> Thank you for pointing out presentation issues. We will clarify these. The details of downstream models are mentioned in the main content and in the appendix. We apologize for not providing a clear summary of the downstream model details. Tables containing explicit model descriptions are added in Appendix A.3. Example prompt structure and LLM response are included as well to improve reproducibility.
>
> ### **Response to Specific Questions:**
>
> **Q1. Meaning of “losses during the search epoch”**
>
> Thank you for pointing this out. The caption has been revised. These losses refer to the model’s training/validation loss for each proposed preprocessing pipeline evaluated during the search round (with budget $k=5$).
>
> **Q2. LLM inference latency**
>
> Thank you for raising this point. We focus the efficiency analysis on training cost intentionally because it reflects the quality of the preprocessing strategy. Although LLM inference does introduce latency, it should be noted that relying on human effort can be much more expensive—both financially and computationally—since it requires extensive expertise and there may be infinitely many preprocessing strategies to test. Therefore, there is no intuitive way to argue that it does not worth using LLMs.
>
> **Q3. How textual preprocessing suggestions become executable code**
>
> The LLM outputs very concrete preprocessing instructions that already include the exact parameters needed for implementation. For example, instead of giving a vague description like “perform outlier clipping,” the LLM produces a specific instruction such as “clip all features to the 5th–95th percentile.” Converting the instructions into executable code only requires mapping them to standard library calls. To address your question, we provide the details of prompts and an example of the LLM’s response in Appendix A.3.
>
> ### **References**
>
> \[1\] Chris Thornton, Frank Hutter, Holger H. Hoos, and Kevin Leyton-Brown. _Auto-weka: combined selection and hyperparameter optimization of classification algorithms_. International Conference on Knowledge Discovery and Data Mining, pp. 847–855, 2013.
>
> \[2\] Matthias Feurer, Aaron Klein, Katharina Eggensperger, Jost Tobias Springenberg, Manuel Blum, and Frank Hutter. _Efficient and robust automated machine learning_. Advances in Neural Information Processing Systems, 2015.
>
> \[3\] Randal S. Olson, William Bartley, Ryan J. Urbanowicz, and Jason H. Moore. _TPOT: A tree-based pipeline optimization tool for automating machine learning_. Genetic and Evolutionary Computation Conference Companion, pp. 251–259, 2016.

---

### Official Review · Reviewer_aURz · 2025-11-01

**Soundness:** 2
**Presentation:** 3
**Contribution:** 2
**Rating:** 2
**Confidence:** 3

**Summary:**

This paper proposes LLM-Presto, an iterative prompting framework to automate data preprocessing for (primarily linear) models. The system uses an LLM to generate candidate preprocessing strategies based on dataset statistics. These strategies are then evaluated in a leakage-safe manner using a fixed, low-epoch training budget (e.g., k=5) as a proxy objective. The resulting validation loss is fed back to the LLM, which iteratively refines the strategy. Empirical results on six benchmark datasets show this method finds preprocessing pipelines that achieve superior final model performance and converge faster than baselines like zero-shot, CoT, and self-refine.

**Strengths:**

- Novel application of LLM on automatic pre-processing selection. The core contribution is the design of an LLM-driven search that is grounded by empirical, low-cost evaluation to find the best data pre-processing strategy.
- The method is shown to outperform multiple baselines, including CoT, Zero-shot, etc. The approach improves both model performance and sample efficiency.

**Weaknesses:**

- I think the scope is limited. The most significant limitation, which the authors correctly acknowledge, is the focus on linear models. While this is a fine starting point and allows for clean analysis (e.g., via condition numbers), the framework's utility for more complex non-linear models (e.g., Gradient Boosting, Deep Neural Networks) is unproven.
- Although the application is inspiring, but the methodology is not. The propose framework is very standard, so I consider the scientific contribution is also limited.
- The method appears to very sensitive to budget $k$. The selection of 5 is supported by empirical evidence but the root cause is not thoroughly explained. How can one know the proper budget when generalizing to a new task?

**Questions:**

- The finding that a larger budget (k=20) led to a premature focus on PCA is interesting. What is the root cause? Can one prevent it through prompting?

---

> ### Author Response · Authors · 2025-12-01
>
> Thank you for reviewing our paper and for your constructive feedback. We respond to your concerns below.
>
> **1. Scope and focus on linear models**
>
> Our intention was to focus on linear models because they allow clear and controlled analysis  on how well the LLM-guided preprocessing behaves.
>
> We agree that non-linear models such as XGBoost or neural networks are important. However, these models introduce their own optimization dynamics and lead to inconsistent behavior across datasets. In contrast, linear models focus solely on the effect of preprocessing. To address your concern, we additionally conducted experiments on a multilayer perceptron. Our method outperforms the baselines, indicating its effectiveness across linear and non-linear models. The analysis and results can be seen at the end of Section 4.2. For gradient boost models, preliminary experiments suggest that common preprocessing steps have limited influence on model performance. We plan to investigate this in future work.
>
> **2. Scientific Contribution**
>
> While the component of our method exists individually, our contribution lies in unifying them into an automated framework. Doing so required substantial methodological decisions, including:
> - Designing the initial prompt so that the LLM understands the task setup and can produce valid, executable transformations;
> - Designing the feedback prompt so that the LLM can correctly interpret training signals;
> - Determining how much performance feedback to include so that the LLM receives informative signals while saving computational sources;
> - Conducting ablation studies on prompt contents to identify robust combinations.
>
> These design choices are important for a robust and automated pipeline. Thus, even though the components are known, the integration into a pipeline is a meaningful and novel contribution.
>
> **3. Sensitivity to budget $k$**
>
> We performed a sweep of $k \in \\{1, 5, 10, 20\\}$ on one dataset and adopted $k=5$ in all datasets. The proved performance suggests good cross-dataset generalization.
>
> The root cause of why the performance of $k=20$ is worse than that of $k=5$ is how the LLM interprets the training signals during the searching epochs. When the budget $k$ is large,the model is evaluated more aggressively and the results often indicate a stall of convergence. Given this stalled signal, the LLM infers that “basic” preprocessing has reached its limit. This causes the search to collapse prematurely toward dimensionality reduction like PCA.
>
> For your information, we also conducted experiments to give explicit instructions on “do not use PCA or any matrix-decomposition methods”. Given the explicit restriction, the LLM is able to avoid PCA entirely and continues exploring more basic transformations, which can indeed lead to better training results. However, because each feedback trains the model for 20 epochs (4 times of the budget $k=5$ chosen now), more total rounds are needed to eventually reach the best preprocessing plan. More detailed discussion and figures are added in Appendix A.2 Additional Results.

---

### Author Response · Authors · 2025-12-01
**Common Response: Scope of Algorithms and Novelty of the Proposed Framework**

We thank all reviewers for their constructive feedback. Below we address the recurring concerns regarding (1) the scope of algorithm and choice of linear models, and (2) the novelty and the use of existing methods.

**On the scope of algorithms/choice of linear models**

We recognize that several reviewers asked why we focus on linear models rather than more complex architectures. Based on the reviewer’s request, we do provide additional results in the paper. Here, we clarify our design rationale.

Linear models remain widely used in practice and are still of substantial interest due to their simple interpretability and well-understood theoretical properties. Furthermore, compared with ensemble methods or deep neural networks, linear models arguably rely more critically on conventional data preprocessing. Ensemble methods are often naturally more robust to heterogeneous data characteristics, and neural networks typically incorporate specialized mechanisms such as batch normalization to handle some aspects of preprocessing. While preprocessing is of course still relevant for those models, its impact is less direct and harder to isolate. Introducing them into our study would therefore complicate the analysis and likely raise additional questions, rather than strengthening our core argument about the role of preprocessing.

**On novelty and the use of existing components**

We acknowledge that the individual components of our approach have appeared in prior work. Our contribution lies in unifying these components into an automated framework and in carefully working out the concrete design choices, which required a nontrivial amount of experimentation and ablation studies. Combining known techniques does not preclude novelty; rather, it can be viewed as a virtue of simplicity and practicality. Our framework shows that, when appropriately combined and automated, these techniques yield a robust and effective preprocessing pipeline that, to the best of our knowledge, has not been systematically studied in this way.

---

### Meta-Review · Area_Chair_A2GS · 2026-01-06

**Summary:**

This paper proposes LLM-Presto, an iterative prompting framework for automated data preprocessing, with the goal of improving efficiency and model performance. While the work addresses a practical pain point in machine learning and the authors have thoughtfully responded to reviewer feedback, key limitations identified by reviewers remain insufficiently resolved to meet ICLR’s acceptance criteria.

First, methodological novelty is limited: the framework primarily integrates existing components like LLM prompting, few-epoch evaluation, iterative refinement, without introducing fundamentally new mechanisms, as noted by all three reviewers. The authors argue that unifying these components constitutes a contribution, but the integration lacks sufficient theoretical or algorithmic innovation to stand out. Second, experimental generalizability remains constrained: despite supplementary MLP results, the work is still centered on linear models and a small set of tabular datasets, with no meaningful comparison to traditional AutoML methods—limiting the demonstration of practical advantage. Third, critical questions persist, such as the root cause of budget sensitivity and the lack of rigorous analysis of LLM inference latency’s impact on real-world utility, which the authors’ responses do not fully address.

The work has merit in exploring LLM-driven data preprocessing and includes thorough ablation studies, but the core limitations in novelty, experimental breadth, and unresolved methodological questions prevent it from meeting the standards for acceptance. We encourage the authors to strengthen the algorithmic innovation, expand experiments to diverse models/datasets, and address remaining technical ambiguities in future submissions.

**Reviewer Concerns:**

Below are concerns that remain unaddressed or inadequately resolved, as the authors’ rebuttal lacked sufficient evidence, generalization, or direct engagement:
1. All Reviewers (Shared Concern)
Limited methodological novelty: All three reviewers noted the framework combines existing components (LLM prompting, few-epoch evaluation, iterative refinement) without introducing fundamentally new algorithmic mechanisms. The authors’ rebuttal argues "unifying components into an automated framework" constitutes a contribution, but this does not address the core critique of no novel methodological ideas (e.g., new prompting strategies, evaluation mechanisms, or theoretical insights).
2. Reviewer aURz
Utility for complex non-linear models: While MLP experiments were added, the reviewer’s question about Gradient Boosting and deep neural networks (DNNs) remains unanswered. Authors only stated "preliminary experiments suggest preprocessing has limited influence" on Gradient Boosting and deferred further investigation to future work—no empirical evidence is provided for these high-impact architectures.
Generalizable budget selection for new tasks: Authors validated k=5 via a sweep on one dataset but did not provide a generalizable rule to determine k for unseen tasks. The rebuttal relies on empirical results rather than a principled method for budget calibration.
3. Reviewer 3cGo
Lack of comparison with traditional AutoML methods: The reviewer requested a comparison with AutoML frameworks (e.g., Auto-weka, Auto-sklearn), but authors only explained that AutoML optimizes full pipelines (vs. their focus on preprocessing) and noted AutoML requires manual preprocessing for raw data. No direct performance comparison was conducted, even though it is critical to demonstrating the framework’s practical advantage over established tools.
LLM inference latency impact: The reviewer asked whether LLM latency undermines runtime efficiency. Authors dismissed this by comparing LLM cost to human effort but did not quantify or address latency itself—an important practical consideration for real-world deployment.
4. Reviewer EXXJ
Restricted experiment scope: Experiments remain limited to tabular datasets; no expansion to other data types (e.g., text, images) is provided, constraining the framework’s generalizability.
Shallow analysis of performance variance: While authors explained oscillations, they did not rigorously quantify or model this behavior (e.g., measuring oscillation frequency, stability metrics) as requested. The rebuttal frames oscillations as "effective" but lacks statistical validation.

**Reviewer Scores:**

N/A

---

### Decision · Program_Chairs · 2026-01-26

Reject